 SciPost Phys. Lect. Notes 79 (2024)

# Introduction to dynamical mean-field theory of randomly connected neural networks with bidirectionally correlated couplings

**Wenxuan Zou[1,2] and Haiping Huang[1,3]⋆**

**1** PMI Lab, School of Physics, Sun Yat-sen University,
Guangzhou 510275, People's Republic of China
**2** Department of Physics, Duke University, Durham, North Carolina, United States of America
**3** Guangdong Provincial Key Laboratory of Magnetoelectric Physics and Devices,
Sun Yat-sen University, Guangzhou 510275, People's Republic of China

⋆ huanghp7@mail.sysu.edu.cn

## Abstract

Dynamical mean-field theory is a powerful physics tool used to analyze the typical behavior of neural networks, where neurons can be recurrently connected, or multiple layers of neurons can be stacked. However, it is not easy for beginners to access the essence of this tool and the underlying physics. Here, we give a pedagogical introduction of this method in a particular example of random neural networks, where neurons are randomly and fully connected by correlated synapses and therefore the network exhibits rich emergent collective dynamics. We also review related past and recent important works applying this tool. In addition, a physically transparent and alternative method, namely the dynamical cavity method, is also introduced to derive exactly the same results. The numerical implementation of solving the integro-differential mean-field equations is also detailed, with an illustration of exploring the fluctuation dissipation theorem.

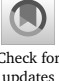

# 1   Introduction

Stochastic differential equations (SDEs) have a very wide range of applications in physics [1], biology, neuroscience and machine learning (see many examples in understanding the brain [2]). Recently, as the world-wide brain projects are being promoted, and the artificial intelligence starts the fourth industrial revolution, understanding how cognition arises both for a natural brain or an artificial algorithm (like Chat GPT [3]) becomes increasingly important. Better understanding leads to precise predictions, which is impossible without solid mathematical foundation. Stochastic noise inherent in the neural dynamics can either stem from the algorithm details (e.g, in stochastic gradient descent of a deep learning cost function [4], the noise is anisotropic and changes with the training) or stem from unreliable synaptic transmission and noisy background inputs [5]). Therefore, SDEs provide a standard tool to model the complex dynamics common in complex systems.

In particular, one can write the SDE into a Langevin dynamics equation, and put the dynamics into the seminal Onsager-Machlup formalism [6], which introduces the concept of action in a path-integral framework [7]. However, the form of the Onsager-Machlup action is not amenable for calculating the quenched disorder average as commonly required in studying typical behaviors of random neural networks. To overcome this drawback, a response field is introduced and thus the correlation (spontaneous fluctuation) as well as the response function can be easily derived (see below). This new field-theoretical approach is called the Martin-Sigga-Rose-De Dominics-Janssen (MSRDJ) formalism [8–10]. The MSRDJ formalism has been used to analyze the recurrent neural networks, e.g., studying the onset of chaos [11–13], and to analyze deep neural networks in recent years [14–16]. In particular, a dynamical partition function was derived for dynamical weight variables in supervised learning, from which the correlation and response functions are computed [17], while a recent work derived dynamical field theory for infinitely wide neural networks trained with gradient flow, at an expensive computational cost [15]. We will next provide a detailed explanation of this field-theoretical formalism, yet in random neural networks with predesigned synaptic structures.

The MSRDJ formalism bears concise mathematics, resulting in the mean-field description of the original high dimensional stochastic dynamics. The same equation can also be derived using a physically more transparent method, namely the dynamical cavity method first introduced in the classic book [18] and then reused in specific contexts [19–22]. In essence, a neuron is added into the system, and the associated impacts on other neurons are seft-consistently derived using the linear response approximation. The dynamics of this additional neuron bears the same characteristics with other neurons, thereby being a representative of the original high dimensional dynamics. In addition, the static version of this method can be used to derive the fixed point solution of the dynamics [21, 22]. By numerically solving the intergro-differential equations involving the correlation and response functions, one can fur-

ther probe the fundamental fluctuation-dissipation relation in equilibrium dynamics, which we shall show in the last section of this tutorial. All technical details are given in the Appendices.

We remark that the MSRDJ formalism has been introduced in several nice review papers [23–25] and a recent book [7]. The review paper in 2005 [23] focuses on $p$-spin spherical model when introducing the formalism, while the review paper in 2015 [24] gives a very nice introduction of the formalism and the associated Feynman diagrams as a tool for carrying out perturbative expansions of the action, but only single variable stochastic dynamics are considered. The recent review paper [25] also gives a very nice introduction of the MSRDJ formalism and the Feynman diagrammatic expansion, and supersymmetric formulation of the stochastic dynamics as well. The recent book [7] also gives a pedagogical description of the formalism and applications to stochastic dynamics. However, the random neural network we consider here is not covered in these reviews and the recent book [7]. Furthermore, the powerful dynamic cavity method based on transparent physics intuition was also not covered in previous review papers. We also discuss dynamics fixed-point analysis and the fluctuation dissipation theorem, which thus provides necessary tools and concepts in analyzing the relationship between response and correlation in non-equilibrium systems. Some recent progresses applying this formalism are also included. We expect this tutorial will benefit beginners interested in this interdisciplinary field. For tutorial purpose, we do not plan to cite all papers related to the MSRDJ formalism, but rather, only the first papers related to introduced concepts or methods, or relevant reviews are cited.

## 2 Random neural networks with bidirectionally correlation synapses

We consider a random neural network composed of $N$ fully-connected neurons. The state of each neuron in time $t$ is characterized by the synaptic current $x_i(t)$, $i = 1, \ldots, N$, which obeys the following non-linear dynamical equation,

$$\frac{dx_i(t)}{dt} = -x_i(t) + g \sum_{j=1}^{N} J_{ij} \phi_j(t) + \sigma \xi_i(t), \tag{1}$$

where $g$ is the coupling strength and $\phi_j(t) = \phi(x_j(t))$ is the transfer function that transforms current to firing rate. A Gaussian white-noise $\xi_i(t)$ of zero mean and unit variance $\langle \xi_i(t)\xi_j(t') \rangle = \delta_{ij}\delta(t-t')$ is introduced to model the stochastic nature of external sensory inputs (e.g., in cortical circuits [2]). The parameter $\sigma$ serves as the noise strength. Each element $J_{ij}$ of the connection matrix is drawn from a Gaussian distribution with zero mean and variance $\overline{J_{ij}^2} = 1/N$. In fact, $J_{ij}$ may correlate with $J_{ji}$ for each pair of neurons. This asymmetric correlation is thus characterized as follows,

$$\overline{J_{ij}J_{ji}} = \frac{\eta}{N}, \tag{2}$$

where $\eta \in [-1, +1]$ describes the degree of asymmetry. In particular, the connections are fully symmetric when $\eta = 1$ and fully asymmetric when $\eta = 0$ [26]. Besides, we set $J_{ii} = 0$ to remove all the self-coupling interaction. Note that other types of network structures can also be similarly treated with the following MSRDJ formalism, e.g., the low-rank random recurrent networks [27], and the random recurrent networks with gating mechanisms [28].

The dynamical mean-field theory (DMFT) equation of this model is easy to acquire when the asymmetric correlation is absent i.e., $\eta = 0$ [11, 29]. In this case, when we consider the limit $N \to \infty$, the input current each neuron receives via the coupling $g \sum_{j=1}^{N} J_{ij} \phi_j(t)$

converges to a Gaussian field according to the central limit theorem. Therefore, the complex network dynamics can be simplified to an effective single-neuron dynamics,

$$\dot{x}(t) = -x(t) + \gamma(t), \tag{3}$$

where $\gamma(t)$ is the effective Gaussian noise with covariance $\langle \gamma(t)\gamma(t') \rangle = g^2 C(t,t') + \sigma^2 \delta(t-t')$. The auto-correlation of firing rates $C(t,t')$ is self-consistently defined as

$$C(t,t') = \langle \phi(t)\phi(t') \rangle, \tag{4}$$

thereby closing the DMFT equation. Note that $\overline{\cdot}$ (or $\mathbb{E}[\cdot]$ in the following) and $\langle \cdot \rangle$ represent the quenched-disorder average and thermal average (over different noise trajectories), respectively.

However, when the asymmetric correlation between connections is present, the mean-field description can not be obtained directly from the central limit theorem. This is because in the summation of the afferent currents, a non-negligible correlation between $J_{ij}$ and $\phi_j(t)$ emerges via $J_{ji}$ when $\eta \neq 0$, and thus the central limit theorem breaks down. In the following sections, we introduce two powerful physics tools to tackle this challenge and derive the DMFT equation for random neural networks of an arbitrary asymmetric correlation level.

## 3 Dynamical mean-field theory

### 3.1 Generating functional formalism

We first introduce the generating functional formalism and then unfold the derivation following the standard procedure. The main idea of this method is to recast the original high-dimensional dynamical equation to the path integral formalism. Here, we consider the MSRDJ path integral. The moment-generating functional with a specific action corresponding to the dynamical equation can be written out explicitly, which is helpful to reduce the dynamics to a low-dimensional mean-field description.

First, we add a perturbation $j_i(t)$ to the original dynamical equation [Eq. (1)],

$$\dot{x}_i(t) = -x_i(t) + g \sum_{j=1}^N J_{ij}\phi_j(t) + j_i(t) + \sigma\xi_i(t), \quad i = 1, \dots, N, \tag{5}$$

which will be useful in the following derivation. There are two types of discretization schemes (Ito versus Stratonovich convention). The Ito one is simpler because the equal time response function vanishes. Note that for different times, the two discretization schemes are equivalent [25]. Therefore, we discretize the dynamical equations under the Ito convention,

$$x_i[t] - x_i[t-1] = -x_i[t-1]h + g \sum_{j=1}^N J_{ij}\phi_j[t-1]h + j_i[t-1]h + \sigma\xi_i[t-1], \tag{6}$$

where $h$ is a time interval between two consecutive time steps, and $[t]$ indicates the discrete time index. The white noise $\xi_i[t] = \int_t^{t+h} \xi_i(s)\,\mathrm{d}s$ becomes a Wiener process with the following statistics,

$$\begin{aligned}
\langle \xi_i[t]\xi_j[t'] \rangle &= \int_t^{t+h}\int_{t'}^{t'+h} \langle \xi_i(s)\xi_j(s') \rangle \,\mathrm{d}s\,\mathrm{d}s' \\
&= \int_t^{t+h}\int_{t'}^{t'+h} \sigma^2\delta_{ij}\delta(s-s')\,\mathrm{d}s\,\mathrm{d}s' \\
&= \delta_{ij}\delta_{tt'}\sigma^2 h,
\end{aligned} \tag{7}$$

where $\delta_{ij}$ is a Kronecker delta function. Because of the Markovian property, we introduce the joint distribution of the currents $\{\boldsymbol{x}(t)\}_{t=1}^{T}$ across time and space by Dirac delta functions,

$$P\left(\{\boldsymbol{x}(t)\}_{t=1}^{T}\right) = \int \prod_{i,t} p(\xi_i[t])\,\mathrm{d}\xi_i[t]\,\delta\Big(x_i[t+1] - x_i[t] + x_i[t]h$$
$$- g\sum_{j=1} J_{ij}\phi_j[t]h - j_i[t]h - \sigma\xi_i[t]\Big), \tag{8}$$

where the initial current state $\boldsymbol{x}[0]$ can be arbitrarily chosen, which does not influence the derivation, and $T$ denotes the length of the trajectory.

We next represent these delta functions by their Fourier integral as $\delta(x) = \frac{1}{2\pi i}\int_{-i\infty}^{i\infty} \mathrm{d}\tilde{x}\, e^{-\tilde{x}x}$,

$$P\left(\{\boldsymbol{x}(t)\}_{t=1}^{T}\right) = \prod_{i,t}\left(\int p(\xi_i[t])\,\mathrm{d}\xi_i[t]\int_{-i\infty}^{i\infty}\frac{\mathrm{d}\tilde{x}_i[t]}{2\pi i}\exp\Big[-\tilde{x}_i[t]\Big(x_i[t+1] - x_i[t] + x_i[t]h\right.$$
$$\left. - g\sum_{j=1} J_{ij}\phi_j[t]h - j_i[t]h - \sigma\xi_i[t]\Big)\Big]\right)$$
$$= \prod_{i,t}\left(\int_{-i\infty}^{i\infty}\frac{\mathrm{d}\tilde{x}_i[t]}{2\pi i}\exp\Big[-\tilde{x}_i[t]\Big(x_i[t+1] - x_i[t] + x_i[t]h\right.$$
$$\left. - g\sum_{j=1} J_{ij}\phi_j[t]h - j_i[t]h - \frac{\sigma^2}{2}\tilde{x}_i[t]h\Big)\Big]\right), \tag{9}$$

where Eq. (7) is used to derive the last equality. Hence, we can formally define the moment-generating functional of the stochastic dynamics,

$$Z[\mathrm{j},\tilde{\mathrm{j}}|\mathrm{J}] = \prod_{i,t}\left(\int_{-\infty}^{\infty}\mathrm{d}x_i[t]\exp\left(\tilde{j}_i[t]x_i[t]h\right)\right)P\left(\{\boldsymbol{x}(t)\}_{t=1}^{T}\right)$$
$$= \prod_{i,t}\left(\int_{-\infty}^{\infty}\mathrm{d}x_i[t]\int_{-i\infty}^{i\infty}\frac{\mathrm{d}\tilde{x}_i[t]}{2\pi i}\exp\left(\tilde{j}_i[t]x_i[t]h + j_i[t]\tilde{x}_i[t]h\right)\right)$$
$$\times \exp\left\{\left[-h\sum_{i,t}\tilde{x}_i[t]\left(\frac{x_i[t+1] - x_i[t]}{h} + x_i[t] - g\sum_{j=1} J_{ij}\phi_j[t] - \frac{\sigma^2}{2}\tilde{x}_i[t]\right)\right]\right\}, \tag{10}$$

where $\tilde{\mathrm{j}}$ and $\mathrm{j}$ are two types of source fields, whose physical meaning would be clear below. The source $\mathrm{j}$ could be an external perturbation to which the response is measured by the response field $\tilde{x}$, which allows one to compute the linear response function by taking the correlation with $x$ (see below). Taking the continuous limits of $T \to \infty$ and $h \to 0$ at the same time, we obtain $h\sum_{t=0}^{T} f(t) = \int f(t)\,\mathrm{d}t$, and $\lim_{h\to 0}\frac{x_i[t+1]-x_i[t]}{h} = \dot{x}_i(t)$. We also introduce the notations $\prod_{i,t}\mathrm{d}x_i[t] \overset{h\to 0}{\to} \mathcal{D}\boldsymbol{x}$ and $\prod_{i,t}\frac{\mathrm{d}\tilde{x}_i[t]}{2\pi i} \overset{h\to 0}{\to} \mathcal{D}\tilde{\boldsymbol{x}}$ for simplicity. Under the continuous limit, the moment generating functional reads,

$$Z[\mathrm{j},\tilde{\mathrm{j}}|\mathrm{J}] = \int \mathcal{D}\boldsymbol{x}\mathcal{D}\tilde{\boldsymbol{x}}\exp\left(-S[\boldsymbol{x},\tilde{\boldsymbol{x}}|\mathrm{J}] + \sum_{i=1}^{N}\int \tilde{j}_i(t)x_i(t)\,\mathrm{d}t + \sum_{i=1}^{N}\int j_i(t)\tilde{x}_i(t)\,\mathrm{d}t\right), \tag{11}$$

where the action of the dynamical equation is naturally introduced as,

$$S[\boldsymbol{x},\tilde{\boldsymbol{x}}|\mathrm{J}] = \sum_{i=1}^{N}\int \tilde{x}_i(t)\left(\dot{x}_i(t) + x_i(t) - g\sum_{j=1}^{N} J_{ij}\phi_j(t) - \frac{\sigma^2}{2}\tilde{x}_i(t)\right)\mathrm{d}t. \tag{12}$$

It is easy to verify that when $N \to \infty$, the out-of-equilibrium behavior is independent of the realization of the disorder [30]. We thus focus on the typical behavior of the self-averaging dynamical partition function $Z[j, \tilde{j}|J]$. This partition function is simpler compared to its equilibrium counterpart, as the zero source generating functional is identical to one. In the dynamical setting, taking the average of $Z[j, \tilde{j}|J]$ over $P(J)$ is sufficient to get the thermal and disorder averaged two-point functions, such as correlation and response. This is in contrast to the equilibrium spin glass theory where a replica trick is commonly applied to obtain the disorder anverage of the free energy function [29]. In fact, computing the average of $\mathbb{E}_J Z[j, \tilde{j}|J]$ reduces to computing $\mathbb{E}_J \exp(-S[\boldsymbol{x}, \tilde{\boldsymbol{x}}|J])$. To proceed, we decompose the connection into symmetric and asymmetric parts [26],

$$J_{ij} = J_{ij}^s + k J_{ij}^a, \tag{13}$$

where $J_{ij}^s = J_{ji}^s$ and $J_{ij}^a = -J_{ji}^a$, both of which follow the centered Gaussian distribution with the same variance,

$$\overline{J_{ij}^s J_{ij}^s} = \overline{J_{ij}^a J_{ij}^a} = \frac{1}{N} \frac{1}{1+k^2}. \tag{14}$$

Under this decomposition, it is easy to derive that,

$$\overline{J_{ij} J_{ij}} = \frac{1}{N}, \qquad \overline{J_{ij} J_{ji}} = \frac{1}{N} \frac{1-k^2}{1+k^2}, \tag{15}$$

which gives $k^2 = (1-\eta)/(1+\eta)$.

Now, we can deal with the term involving $J_{ij}$,

$$\sum_{i \neq j} \tilde{x}_i(t) J_{ij} \phi_j(t) = \sum_{i \neq j} \tilde{x}_i(t) \left[ J_{ij}^s + k J_{ij}^a \right] \phi_j(t)$$
$$= \sum_{i < j} \left\{ J_{ij}^s \left[ \tilde{x}_i(t) \phi_j(t) + \tilde{x}_j(t) \phi_i(t) \right] + k J_{ij}^a \left[ \tilde{x}_i(t) \phi_j(t) - \tilde{x}_j(t) \phi_i(t) \right] \right\}. \tag{16}$$

Then, carrying out the average over $J_{ij}^s$ and $J_{ij}^a$ leads to

$$\mathbb{E}_{J^s, J^a} \exp\left( \int dt \sum_{i<j} \left\{ J_{ij}^s \left[ \tilde{x}_i(t) \phi_j(t) + \tilde{x}_j(t) \phi_i(t) \right] + k J_{ij}^a \left[ \tilde{x}_i(t) \phi_j(t) - \tilde{x}_j(t) \phi_i(t) \right] \right\} \right)$$
$$= \exp\left( \frac{g^2}{2N} \sum_{i \neq j} \iint \left\{ \left[ \tilde{x}_i(t) \phi_j(t) \tilde{x}_i(t') \phi_j(t') \right] + \eta \left[ \tilde{x}_i(t) \phi_j(t) \tilde{x}_j(t') \phi_i(t') \right] \right\} dt \, dt' \right)$$
$$\approx \exp\left( \frac{g^2}{2N} \int \int \left[ \sum_i \tilde{x}_i(t) \tilde{x}_i(t') \sum_j \phi_j(t) \phi_j(t') + \eta \sum_i \tilde{x}_i(t) \phi_i(t') \sum_j \phi_j(t) \tilde{x}_j(t') \right] dt \, dt' \right). \tag{17}$$

Note that, we have added back the negligible diagonal term $(i = j)$ to arrive at the last equality. Then, we define $Z[j, \tilde{j}] = \mathbb{E}_J Z[j, \tilde{j}|J]$, the average moment-generating functional is given by

$$Z[j, \tilde{j}] = \int \mathcal{D}\boldsymbol{x} \, \mathcal{D}\tilde{\boldsymbol{x}} \exp\left( -S_0[\boldsymbol{x}, \tilde{\boldsymbol{x}}] + \frac{\sigma^2}{2} \tilde{\boldsymbol{x}} \cdot \tilde{\boldsymbol{x}} + \tilde{j} \cdot \boldsymbol{x} + j \cdot \tilde{\boldsymbol{x}} \right.$$
$$\left. + \frac{g^2}{2N} \int \int \left[ \sum_i \tilde{x}_i(t) \tilde{x}_i(t') \sum_j \phi_j(t) \phi_j(t') + \eta \sum_i \tilde{x}_i(t) \phi_i(t') \sum_j \phi_j(t) \tilde{x}_j(t') \right] dt \, dt' \right), \tag{18}$$

where $S_0[\boldsymbol{x}, \tilde{\boldsymbol{x}}] = \tilde{\boldsymbol{x}} \cdot [\dot{\boldsymbol{x}} + \boldsymbol{x}]$ is called the free action. $\boldsymbol{f} \cdot \boldsymbol{g} = \sum_{i=1}^{N} \int f_i(t) g_i(t) \, \mathrm{d}t$ is introduced for compactness. From Eq. (18), we have to introduce two auxiliary overlaps,

$$
\begin{aligned}
Q_1(t, t') &= \frac{g^2}{N} \sum_j \phi_j(t) \phi_j(t'), \\
Q_2(t, t') &= \frac{g^2 \eta}{N} \sum_j \phi_j(t) \tilde{x}_j(t'),
\end{aligned}
\tag{19}
$$

which converges to (scaled) Gaussian fields due to the central limit theorem when $N$ is sufficiently large. Thus, we can insert these order parameters into Eq. (18) by the Fourier integral representation of Dirac delta functions,

$$
\begin{aligned}
&\delta\left( -\frac{N}{g^2} Q_1(t, t') + \sum_j \phi_j(t) \phi_j(t') \right) \\
&= \frac{1}{2\pi} \int \mathcal{D}\hat{Q}_1(t, t') \exp\left[ \int \int \hat{Q}_1(t, t') \left( -\frac{N}{g^2} Q_1(t, t') + \sum_j \phi_j(t) \phi_j(t') \right) \mathrm{d}t \, \mathrm{d}t' \right], \\
&\delta\left( -\frac{N}{g^2} Q_2(t, t') + \eta \sum_j \phi_j(t) \tilde{x}_j(t') \right) \\
&= \frac{1}{2\pi} \int \mathcal{D}\hat{Q}_2(t, t') \exp\left[ \int \int \hat{Q}_2(t, t') \left( -\frac{N}{g^2} Q_2(t, t') + \eta \sum_j \phi_j(t) \tilde{x}_j(t') \right) \mathrm{d}t \, \mathrm{d}t' \right].
\end{aligned}
\tag{20}
$$

Finally, we can re-express the averaged moment-generating functional as

$$
\begin{aligned}
Z[\mathrm{j}, \tilde{\mathrm{j}}] = \int \int \mathcal{D}\boldsymbol{X} \mathcal{D}\mathcal{Q} \exp\Bigg( &-\frac{N}{g^2} \hat{Q}_1 \cdot Q_1 - \frac{N}{g^2} \hat{Q}_2 \cdot Q_2 - S_0[\boldsymbol{x}, \tilde{\boldsymbol{x}}] + \frac{\sigma^2}{2} \tilde{\boldsymbol{x}} \cdot \tilde{\boldsymbol{x}} + \tilde{\mathrm{j}} \cdot \boldsymbol{x} + \mathrm{j} \cdot \tilde{\boldsymbol{x}} \\
&+ \frac{1}{2} \int \int \sum_j \tilde{x}_j(t) Q_1(t, t') \tilde{x}_j(t') \, \mathrm{d}t \, \mathrm{d}t' + \frac{1}{2} \int \int \sum_j \tilde{x}_j(t) Q_2(t, t') \phi_j(t') \, \mathrm{d}t \, \mathrm{d}t' \\
&+ \int \int \sum_j \phi_j(t) \hat{Q}_1(t, t') \phi_j(t') \, \mathrm{d}t \, \mathrm{d}t' + \eta \int \int \sum_j \phi_j(t) \hat{Q}_2(t, t') \tilde{x}_j(t') \, \mathrm{d}t \, \mathrm{d}t' \Bigg),
\end{aligned}
\tag{21}
$$

where $\mathcal{D}\boldsymbol{X} \equiv \mathcal{D}\boldsymbol{x} \mathcal{D}\tilde{\boldsymbol{x}}$, and $\mathcal{D}\mathcal{Q} \equiv \left( \frac{N}{2\pi g^2} \right)^2 \mathcal{D}Q_1(t, t') \mathcal{D}\hat{Q}_1(t, t') \mathcal{D}Q_2(t, t') \mathcal{D}\hat{Q}_2(t, t')$, and we also introduce new notations,

$$
\begin{aligned}
\hat{Q}_1 \cdot Q_1 &= \int \int \hat{Q}_1(t, t') Q_1(t, t') \, \mathrm{d}t \, \mathrm{d}t', \\
\hat{Q}_2 \cdot Q_2 &= \int \int \hat{Q}_2(t, t') Q_2(t, t') \, \mathrm{d}t \, \mathrm{d}t'.
\end{aligned}
\tag{22}
$$

We can now remark that the averaged moment-generating functional is completely factorized over neurons, which implies that the original complex dynamics with $N$ interacting neurons is captured by a mean-field one-neuron system subject to a correlated Gaussian noise. More

compactly, we recast the averaged moment-generating functional as

$$Z[j, \tilde{j}] = \int \mathcal{DQ} \exp\left(Nf(Q, \hat{Q}, x, \tilde{x})\right),$$

$$f(Q, \hat{Q}, x, \tilde{x}) = -\frac{1}{g^2}\hat{Q}_1 \cdot Q_1 - \frac{1}{g^2}\hat{Q}_2 \cdot Q_2 + \log \bar{Z}[j, \tilde{j}],$$

$$\bar{Z}[j, \tilde{j}] = \int \mathcal{DX} \exp\left(\mathcal{L}(Q, \hat{Q}, x, \tilde{x})\right), \tag{23}$$

$$\mathcal{L}(Q, \hat{Q}, x, \tilde{x}) = -S_0[x, \tilde{x}] + \frac{\sigma^2}{2}\tilde{x} \cdot \tilde{x} + \tilde{j} \cdot x + j \cdot \tilde{x} + \frac{1}{2}\tilde{x}^T Q_1 \tilde{x}$$

$$+ \frac{1}{2}\tilde{x}^T Q_2 \phi + \phi^T \hat{Q}_1 \phi + \eta \phi^T \hat{Q}_2 \tilde{x},$$

where $\bar{Z}[j, \tilde{j}]$ is the effective moment generating functional for one-neuron system, which will be mathematically clear at the end of the derivation. Thus, $x, \tilde{x}, j, \tilde{j}, X$ are the mean-field counterpart of their original meaning in the high dimensional space. Accordingly, we have $f \cdot g = \int f(t)g(t)dt$. In addition, we define the new notation involving $\{Q, \hat{Q}\}$ in terms of the quadratic form as $f^T Q g = \int \int f(t)Q(t, t')g(t) \, dt \, dt'$. In $N \to \infty$, we estimate asymptotically the averaged dynamical partition function by applying the Laplace method,

$$Z[j, \tilde{j}] = \int \mathcal{DQ} \exp\left(Nf(Q, \hat{Q}, x, \tilde{x})\right) \approx \exp\left(Nf(Q^\star, \hat{Q}^\star, x, \tilde{x})\right), \tag{24}$$

where $\{Q^\star, \hat{Q}^\star\}$ maximizes the dynamical action $f$. We thus have,

$$\frac{\delta f(Q, \hat{Q}, x, \tilde{x})}{\delta \hat{Q}_1(t, t')} = 0 \to Q_1^\star(t, t') = g^2 \left\langle \phi(t)\phi(t') \right\rangle_{\mathcal{L}},$$

$$\frac{\delta f(Q, \hat{Q}, x, \tilde{x})}{\delta Q_1(t, t')} = 0 \to \hat{Q}_1^\star(t, t') = \frac{g^2}{2} \left\langle \tilde{x}(t)\tilde{x}(t') \right\rangle_{\mathcal{L}},$$

$$\frac{\delta f(Q, \hat{Q}, x, \tilde{x})}{\delta \hat{Q}_2(t, t')} = 0 \to Q_2^\star(t, t') = g^2\eta \left\langle \phi(t)\tilde{x}(t') \right\rangle_{\mathcal{L}}, \tag{25}$$

$$\frac{\delta f(Q, \hat{Q}, x, \tilde{x})}{\delta Q_2(t, t')} = 0 \to \hat{Q}_2^\star(t, t') = \frac{g^2}{2} \left\langle \tilde{x}(t)\phi(t') \right\rangle_{\mathcal{L}},$$

where

$$\langle \mathcal{O} \rangle_{\mathcal{L}} = \frac{\int \mathcal{O}(X) \exp[\mathcal{L}(X)] \mathcal{DX}}{\bar{Z}[j, \tilde{j}]}. \tag{26}$$

This average can be seen as the dynamical mean field measure provided by $\bar{Z}[j, \tilde{j}]$ in the one-neuron system, in analogy with the replica analysis in equilibrium spin glass theory [18, 29]. In the following text, we will omit the subscript $\mathcal{L}$ for simplicity.

Now we come to the physical meaning of the dynamics order parameters. First, it is easy to find that $Q_1^\star(t, t')$ is related to the auto-correlation function, which is

$$C(t, t') = \frac{1}{N}\sum_i \langle \phi_i(t)\phi_i(t') \rangle \to \langle \phi(t)\phi(t') \rangle, \tag{27}$$

and $Q_1^\star(t, t') = g^2 C(t, t')$. Second, $\hat{Q}_1^\star(t, t')$ will always vanish because $\frac{\delta^n}{\delta j(t_1) \cdots \delta j(t_n)} Z[j, 0]|_{j=0} = 0$ [see Eq. (10)]. In other words, these response fields do not propagate. Finally, $Q_2^\star(t, t')$ and $\hat{Q}_2^\star(t, t')$ bear exactly the same physical meaning, which relates to the response function,

$$R(t, t') = \frac{1}{N}\sum_i \frac{\delta \langle \phi_i(t) \rangle}{\delta j_i(t')}\bigg|_{j=0}, \tag{28}$$

SciPost Phys. Lect. Notes 79 (2024)

where the average is taken under the path probability, i.e.,

$$R(t,t') = \frac{1}{N}\sum_i \left.\frac{\delta\langle\phi_i(t)\rangle}{\delta j_i(t')}\right|_{j=0} = \frac{1}{N}\sum_i \langle\phi_i(t)\tilde{x}_i(t')\rangle \to \langle\phi(t)\tilde{x}(t')\rangle\,. \tag{29}$$

Thus, $Q_2^\star(t,t') = g^2\eta R(t,t')$ and $\hat{Q}_2^\star(t,t') = \frac{g^2}{2}R(t',t)$. Moreover, the response function $R(t,t')$ will vanish once $t < t'$ because of the causality that perturbations in a later time do not affect the present and past states. In addition, the equal time response function $R(t,t)$ also vanishes under the Ito convention [25]. It is now clear that the term $Q_2 \cdot \hat{Q}_2$ vanishes because of the causality and the Ito convention (note that $R(t,t')R(t',t) = 0$). Finally, we achieve the final form of the moment generating functional,

$$Z[j,\tilde{j}] = \prod_i^N \bar{Z}[j,\tilde{j}] = \left(\bar{Z}[j,\tilde{j}]\right)^N,$$
$$\bar{Z}[j,\tilde{j}] \propto \int \mathcal{D}X \exp\left(-S_0[x,\tilde{x}] + \tilde{j}\cdot x + j\cdot\tilde{x} + \frac{1}{2}\tilde{x}^T\Gamma\tilde{x} + \eta g^2\,\tilde{x}^T R\phi\right) \tag{30}$$
$$= \int \mathcal{D}X \exp\left(-S[x,\tilde{x}] + \tilde{j}\cdot x + j\cdot\tilde{x}\right),$$

where $\Gamma(t,t') = g^2 C(t,t') + \sigma^2\delta(t-t')$, and the effective action (decomposed into free and interaction parts) reads,

$$S[x,\tilde{x}] = \tilde{x}\cdot[\dot{x}+x] - \frac{1}{2}\tilde{x}^T\Gamma\tilde{x} - \eta g^2\,\tilde{x}^T R\phi\,. \tag{31}$$

The first line of Eq. (30) clearly illustrates that the $N$-neuron interactive system degrades to $N$ factorized one-neuron effective systems. And equation (31) further suggests that the dynamical mean-field description of the $N$-neuron dynamics exists [from Eq. (12)], i.e.,

$$\dot{x}(t) = -x(t) + \eta g^2 \int_0^t R(t,t')\phi(t')\,\mathrm{d}t' + \gamma(t), \tag{32}$$

where $\langle\gamma(t)\gamma(t')\rangle = g^2 C(t,t') + \sigma^2\delta(t-t')$. When $\eta = 0$, Eq. (32) reduces to Eq. (3). It is interesting that the spatially correlated asymmetric connections between neurons in the $N$-neuron system are transformed into (long-term) integration of dynamics history in the one-neuron (mean-field description) system, which was also clarified in the failure of local chaos hypothesis in discrete dynamics of this type of random neural networks [31]. The underlying physics is more transparent in the cavity framework introduced later.

The MSDRJ formalism allows one to derive integro-differential equations involving response and correlation functions. For example, we could compute the dynamical equations of mean-field correlation function $\Delta(t,t') = \langle x(t)x(t')\rangle$ and response function $\chi(t,t') = \left.\frac{\delta\langle x_i(t)\rangle}{\delta j_i(t')}\right|_{j=0}$ for currents. First, we consider two identities,

$$\frac{\delta x(t')}{\delta\tilde{x}(t)} = 0, \qquad \frac{\delta x(t')}{\delta x(t)} = \delta(t-t')\,. \tag{33}$$

Then, we take the path average over the probability defined by Eq. (30),

$$\left\langle\frac{\delta x(t')}{\delta\tilde{x}(t)}\right\rangle = \int \mathcal{D}X \frac{\delta x(t')}{\delta\tilde{x}(t)}\exp\left(-S[x,\tilde{x}]\right)$$
$$= \int \mathcal{D}X x(t')\frac{\delta S[x,\tilde{x}]}{\delta\tilde{x}(t)}\exp\left(-S[x,\tilde{x}]\right) \tag{34}$$
$$= \left\langle x(t')\left(\dot{x}(t)+x(t)-\int\Gamma(t,s)\tilde{x}(s)\,\mathrm{d}s - \eta g^2\int R(t,s)\phi(s)\,\mathrm{d}s\right)\right\rangle = 0,$$

where the integral by parts is used to derive the second equality. This relation will immediately give rise to,

$$\frac{\partial}{\partial t}\Delta(t,t') = -\Delta(t,t') + g^2 \int_0^{t'} \chi(t',s)C(t,s)ds + \sigma^2\chi(t',t) + \eta g^2 \int_0^{t} R(t,s)\langle x(t')\phi(s)\rangle ds. \tag{35}$$

Similarly, the other identity in Eq. (33) leads to,

$$\frac{\partial}{\partial t}\chi(t,t') = -\chi(t,t') + \delta(t-t') + \eta g^2 \int_{t'}^{t} R(t,s)R(s,t')ds. \tag{36}$$

These integro-differential equations are particularly difficult to solve, e.g., no closed form solutions exist except at $\eta = 0$. In general, a perturbative expansion of the non-linear transfer function may be required [13].

## 3.2 Dynamical cavity approach

In this section, we introduce the dynamical cavity approach [18, 20, 21], which is more physically transparent (like its static counterpart [29]). The dynamical cavity approach gives exactly the same DMFT equation that is based on the moment generating functional method. Our starting point is still the $N$-neuron stochastic dynamics,

$$\dot{x}_i(t) = -x_i(t) + g\sum_{j=1}^{N} J_{ij}\phi_j(t) + j_i(t) + \sigma\xi_i(t), \quad i = 1,\dots,N, \tag{37}$$

where the Gaussian noise $\xi_i(t)$ has the variance $\langle \xi_i(t)\xi_j(t')\rangle = \delta_{ij}\delta(t-t')$. Connections $J_{ij}$ are drawn from the centered Gaussian distribution with the variance $\frac{1}{N}$ as well as the covariance $\overline{J_{ij}J_{ji}} = \frac{\eta}{N}$.

First, we add a new neuron into the original system, such that we have a new synaptic current $x_0(t)$ together with the corresponding connections $(J_{0i}, J_{i0})$, for $i = 1,\dots,N$. As a result, all the neurons in the original system will be affected by this new neuron. We regard this impact as a small perturbation in the large network limit. We can thus apply the linear response theory as follows,

$$\begin{aligned}\tilde{\phi}_i(t) &= \phi_i(t) + \sum_{k=1}^{N} \int_0^t \left.\frac{\delta\phi_i(t)}{\delta j_k(s)}\right|_{j=0} j_k(s)\,ds \\ &= \phi_i(t) + \sum_{k=1}^{N} \int_0^t R_{ik}(t,s)[gJ_{k0}\phi_0(s)]\,ds,\end{aligned} \tag{38}$$

where $R_{ik}(t,s) = \left.\frac{\delta\phi_i(t)}{\delta j_k(s)}\right|_{j=0}$ defines the linear response function, and the small perturbation is given by $j_k(s) = gJ_{k0}\phi_0(s)$. Then, we can write down the dynamical equation of $x_0(t)$,

$$\begin{aligned}\dot{x}_0(t) &= -x_0(t) + g\sum_{j\neq 0} J_{0j}\tilde{\phi}_j(t) + j_0(t) + \sigma\xi_0(t) \\ &= -x_0(t) + g\sum_{j=1}^{N} J_{0j}\left[\phi_j(t) + \sum_{k=1}^{N}\int_0^t R_{jk}(t,s)[gJ_{k0}\phi_0(s)]\,ds\right] + j_0(t) + \sigma\xi_0(t) \\ &= -x_0(t) + g\sum_{j=1}^{N} J_{0j}\phi_j(t) + \sigma\xi_0(t) + g^2\int_0^t \sum_{jk} J_{0j}R_{jk}(t,s)J_{k0}\phi_0(s)\,ds + j_0(t),\end{aligned} \tag{39}$$

where the fourth term captures how the asymmetric correlation affects the current state of the new neuron through the response function. The bare field without the effects of synaptic correlation is separated out as follows,

$$\gamma_0(t) = g\sum_{j=1}^{N} J_{0j}\phi_j(t) + \sigma\xi_0(t), \tag{40}$$

which becomes the centered Gaussian field whose variance is given by

$$\langle\gamma_0(t)\gamma_0(t')\rangle = g^2 C(t,t') + \sigma^2\delta(t-t'), \tag{41}$$

where $C(t,t') = \frac{1}{N}\sum_j \phi_j(t)\phi_j(t')$ is the population averaged auto-correlation function.

The computation of the fourth term in Eq. (39) requires us to estimate $\sum_{jk} J_{0j}R_{jk}(t,s)J_{k0}$ which is subject to central limit theorem by construction. We first consider the diagonal part $\sum_j J_{0j}R_{jj}(t,s)J_{j0}$, which will converge asymptotically to its mean because of the negligible variance (of the order $1/\sqrt{N}$),

$$\mathbb{E}_J \sum_j J_{0j}R_{jj}(t,s)J_{j0} = \frac{\eta}{N}\sum_j R_{jj}(t,s). \tag{42}$$

Then, we turn to the non-diagonal part $\sum_{j\neq k} J_{0j}R_{jk}(t,s)J_{k0}$, whose mean is zero due to $\overline{J_{0j}J_{k0}} = 0$, and we should thus consider the fluctuation given by

$$\mathbb{E}_J \sum_{j\neq k}\sum_{j'\neq k'} J_{0j}J_{0j'}J_{k0}J_{k'0}R_{jk}(t,s)R_{j'k'}(t,s) = \sum_{j\neq k}\overline{J_{0j}^2 J_{k0}^2}R_{jk}^2(t,s) \sim \frac{1}{N}, \tag{43}$$

where we assume that $R_{jk}(t,s)$ is of the order $\mathcal{O}(1/\sqrt{N})$ for $j\neq k$, as in the equilibrium limit, the response function has the exactly the same magnitude order with the correlation function ($\mathcal{O}(1/\sqrt{N})$ in fully connected mean-field models), and a proof for a dynamical system is shown in Ref. [21]. Therefore, the contribution from the non-diagonal part can be neglected when $N$ is large, and the dynamical equation of $x_0(t)$ is thus simplified as,

$$\dot{x}_0(t) = -x_0(t) + \gamma(t) + g^2\eta\int_0^t R(t,s)\phi_0(s)\,\mathrm{d}s + j_0(t), \tag{44}$$

where the population averaged response function $R(t,s) = \frac{1}{N}\sum_i \left.\frac{\delta\phi_i(t)}{\delta j_i(s)}\right|_{j=0}$.

The added neuron $x_0(t)$ is not special, and its dynamics is a representative of the typical behavior of other neurons. Therefore, we could omit the subscript 0, and write down the mean-field dynamics as follows,

$$\dot{x}(t) = -x(t) + \gamma(t) + g^2\eta\int_0^t R(t,s)\phi(s)\,\mathrm{d}s, \tag{45}$$

where $\gamma(t)$ is the effective noise with the temporally correlated variance,

$$\langle\gamma(t)\gamma(t')\rangle = g^2 C(t,t') + \sigma^2\delta(t-t'). \tag{46}$$

Finally, to close this self-consistent equation, we further assume that in the large $N$ limit, the population average of the correlation and response functions converge to their path average (with respect to the noise trajectories and the initial conditions). More precisely,

$$C(t,t') = \frac{1}{N}\sum_j \phi_j(t)\phi_j(t') = \langle\phi(t)\phi(t')\rangle,$$
$$R(t,t') = \frac{1}{N}\sum_i \left.\frac{\delta\phi_i(t)}{\delta j_i(t')}\right|_{j=0} = \left\langle\frac{\delta\phi(t)}{\delta j(t')}\right\rangle\bigg|_{j=0}, \tag{47}$$

which leads to the same Eq. (32) that has been previously derived from the moment generating functional. Note that the last equality in the above response function is equivalent to the definition in Eq. (28) [23].

## 4 Numerical and theoretical analysis

### 4.1 Numerical solution of the DMFT equations

In general, the DMFT equation does not have a closed-form solution (e.g., for $\eta \neq 0$). Therefore, we have to solve the equation numerically. In fact, solving the self-consistent DMFT equation [Eq. (32)] is more challenging compared to the counterpart of an equilibrium system. The main reason is that the self-consistent iteration involves time-dependent function ($C(t, t')$ and $R(t, t')$) rather than scalar variables like overlaps in spin glass theory. Note that in mean-field spin glass models [32], the long time-difference limit of the two-point correlation corresponds to the Edwards-Anderson order parameter. Following the previous works [21,33], we give the numerical implementation details of solving the DMFT equations for the current random neural networks below. Codes are available at the Github link [34].

The iteration scheme works in discrete time steps, and we must set a duration $L(ms)$ as well as a time interval $\Delta t(ms)$ for discretization. The time is measured in units of millisecond, which is common in simulating neural dynamics in brain circuits (e.g., time scales for rate dynamics can be considered in this time unit) [2]. In the beginning of the iteration, we initialize the self-consistent function matrix $C[t, t']$ and $R[t, t']$, whose dimensions are both $(L/\Delta t) \times (L/\Delta t)$. In each iteration, we carry out the following steps:

1. Draw $M$ samples of noise trajectories $\{\gamma^a[t]\}_{a=1}^M$ from the multivariate Gaussian distribution $\mathcal{N}(0, g^2 C[t, t'] + \sigma^2/\Delta t)$, where the emergence of $\Delta t$ is the result of discretization for the Dirac delta function.

2. For these noise trajectories, run $M$ corresponding current trajectories independently by a direct discretization,

$$x^a[t+1] = (1 - \Delta t)x^a[t] + \gamma^a[t]\Delta t + g^2 \eta \Delta t^2 \sum_{s=0}^{t} R[t, s]\phi^a[s]. \qquad (48)$$

3. Calculate the self-consistent functions, in which the auto-correlation function $C[t, t']$ is calculated by $1/M \sum_a \phi_a[t]\phi_a[t']$ and the response function is calculated by integrating the dynamics Eq. (36),

$$\chi^a[t+1, t'] = (1 - \Delta t)\chi^a[t, t'] + \delta_{t,t'} + \eta g^2 \Delta t^2 \sum_{s=t'}^{t} R_{\text{iter}}[t, s]R^a[s, t'], \qquad (49)$$

where the superscript $a$ is the trajectory index, and the response function is computed by $R[t, t'] = \phi'(x[t])\chi[t, t']$. Here $R_{\text{iter}}[t, s]$ refers to the response function estimated from the *last iteration step*. After running the dynamics, we compute the new response function $\chi[t, t'] = 1/M \sum_a \chi^a[t, t']$.

We remark that a damping term would be helpful to speed up the convergence and running in parallel several stochastic trajectories is also a useful strategy. An alternative way to compute the response function is using the Novikov's theorem (see Appendix A) [35]. We do not apply this formula in the iteration, as it needs much more trajectories for the convergence.

We compare the observables obtained from the direct simulation of $N$-neuron dynamics [Eq. (1)] and the mean-field solution for the one-neuron dynamics [Eq .(32)] to check the

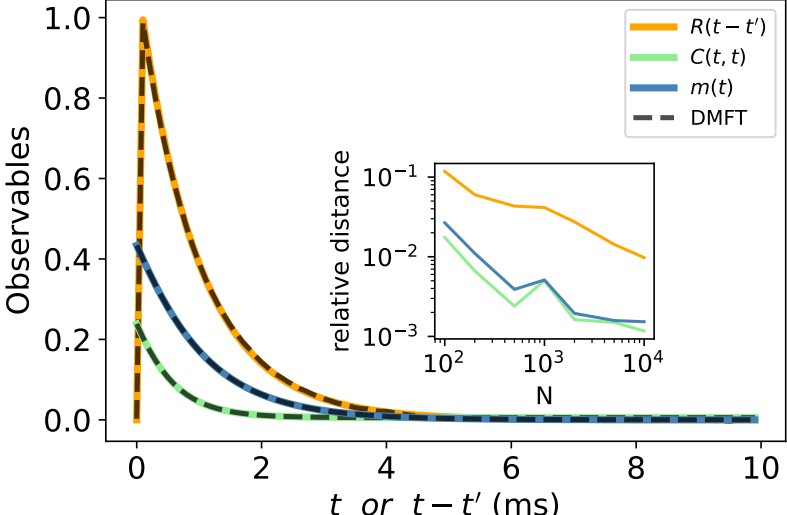

Figure 1: Comparison between observables obtained from direct simulation (color curves) and iterative mean-field solution (dash curves). The parameters set $\{g, \eta, \sigma, \phi, \Delta t, t'\}$ is $\{0.2, 0.5, 0.1, \tanh, 0.1(ms), 10(ms)\}$. Inset: relative differences vs $N$.

effectiveness of the DMFT. Besides the correlation function and response function, we also compare the observable of mean firing-rate $m(t) = \langle \phi(t) \rangle$, which is also an important quantity of the current system. We used Novikov's theorem to compute the response function for this example (see Eq. (A.5) in Appendix A). The curves show an excellent agreement for each observable (Figure. 1), where the curve for the temporal integration is computed from 100 independent runs. For the response function, the argument is selected to be the time difference, as $R(t - t')$ is our focus in the steady state where the time-translation invariance holds. Note that $R(0) = 0$ is a direct result of the Ito convention. The inset shows the relative differences between two types of observables, computed by,

$$\frac{\|\hat{m} - m\|_2}{\|m\|_2}, \qquad \frac{\|\hat{C} - C\|_F}{\|C\|_F}, \qquad \frac{\|\hat{R} - R\|_F}{\|R\|_F}, \tag{50}$$

where hated variables represent the simulation results, while the non-hated ones are DMFT results, and the subscripts 2 and $F$ means the $\ell_2$ norm and Frobenius norm respectively. The relative differences decrease as $N$ grows, which meets our expectation that the DMFT equation predicts the typical behavior of the dynamics under the large network limit.

## 4.2 Analysis of fixed point solutions

In this section, we derive the fixed point solution of the DMFT equation. Under a special choice of model parameters, we could even obtain the analytic fixed point solution in the mean-field description. We then focus on the noise-free case of $\sigma = 0$ with the dynamics:

$$\dot{x}(t) = -x(t) + \hat{\gamma}(t) + g^2 \eta \int_0^t R(t, t') \phi(t') \, dt' + j(t), \tag{51}$$

where $j(t)$ is a perturbation and $\hat{\gamma}(t)$ is the noise-free effective field with the variance,

$$\langle \hat{\gamma}(t) \hat{\gamma}(t') \rangle = g^2 C(t, t'). \tag{52}$$

We assume that the dynamics converges to a fixed point ($\dot{x}(t) = 0$). In the steady state, we get $R(t, t') = R(t - t')$ to simplify the integral term,

$$\int_0^t R(t, t')\phi(t')\,\mathrm{d}t' = \int_0^t R(u)\phi(t-u)\,\mathrm{d}u \overset{t\to\infty}{\longrightarrow} \int_0^\infty R(u)\phi(\infty)\,\mathrm{d}u = R_{\text{int}}\phi^*, \qquad (53)$$

where $R_{\text{int}} = \int_0^\infty R(u)\,\mathrm{d}u$ is the integrated response function and $*$ indicate the steady state. Then, we could obtain the fixed point relation,

$$x^* = \hat{\gamma}^* + w\phi^* + j, \qquad (54)$$

where $w = g^2\eta R_{\text{int}}$, and

$$\langle(\hat{\gamma}^*)^2\rangle = g^2 C. \qquad (55)$$

However, the fixed point relation is not closed, and we must evaluate $R_{\text{int}}$ by its definition. In essence, the integrated response function $R_{\text{int}}$ could be computed by $\langle\frac{\delta\phi^*}{\delta j}\rangle$, setting $j$ to zero later. An equivalent derivation is given in the Appendix B. One can generate a series of noise samples $\hat{\gamma}^*$ from $\mathcal{N}(0, g^2 C^{\text{iter}})$, where the superscript iter denotes the value from the last iteration step. Second, the new observables from these samples are computed as,

$$C = \langle(\phi^*)^2\rangle_{\hat{\gamma}^*}, \qquad R_{\text{int}} = \langle\phi'[\hat{\gamma}^* + w\phi^*][1 + wR_{\text{int}}^{\text{iter}}]\rangle_{\hat{\gamma}^*}. \qquad (56)$$

These equations can be iteratively solved, requiring a high computational complexity due to the noise sampling and fixed point searching for each noise sample.

To achieve an analytic fixed point solution, we choose the ReLU function $\phi(x) = x\Theta(x)$, which is commonly used in machine learning and theoretical neuroscience studies. We could thus recast Eq. (54) to the following form,

$$\phi^* = \phi(\hat{\gamma}^* + w\phi^*), \qquad (57)$$

where $j$ is erased. With the help of ReLU function, we can write $\phi^*$ as a function of $\hat{\gamma}^*$,

$$\phi^* = \frac{\hat{\gamma}^*\Theta(\hat{\gamma}^*)}{1-w} \equiv \psi(\hat{\gamma}^*), \qquad (58)$$

and the response function becomes,

$$R_{\text{int}} = \left\langle\frac{\delta\phi^*}{\delta j}\right\rangle = \left\langle\frac{\delta\phi^*}{\delta\hat{\gamma}^*}\right\rangle = \left\langle\frac{\Theta(\hat{\gamma}^*)}{1-w}\right\rangle. \qquad (59)$$

Therefore, we can derive the self-consistent equations of $C, R_{\text{int}}$ as well as $m$,

$$C = \langle(\phi^*)^2\rangle = \int\left(\frac{\hat{\gamma}^*\Theta(\hat{\gamma}^*)}{1-w}\right)^2 p(\hat{\gamma}^*)\,\mathrm{d}\hat{\gamma}^* = \frac{g^2 C}{2(1-w)^2},$$

$$R_{\text{int}} = \left\langle\frac{\Theta(\hat{\gamma}^*)}{1-w}\right\rangle = \int\frac{\Theta(\hat{\gamma}^*)}{1-w}p(\hat{\gamma}^*)\,\mathrm{d}\hat{\gamma}^* = \frac{1}{2(1-w)}, \qquad (60)$$

$$m = \langle\phi^*\rangle = \int\frac{\hat{\gamma}^*\Theta(\hat{\gamma}^*)}{1-w}p(\hat{\gamma}^*)\,\mathrm{d}\hat{\gamma}^* = \frac{g^2 C}{\sqrt{2\pi}(1-w)},$$

where the integral range covers the entire real value region, and $w = g^2\eta R_{\text{int}}$. In the following, we omit the subscript of $R_{\text{int}}$. These relations will give the analytic fixed point (observables),

$$m = 0, \qquad C = 0, \qquad R = \frac{1 - \sqrt{1 - 2g^2\eta}}{2g^2\eta}. \qquad (61)$$

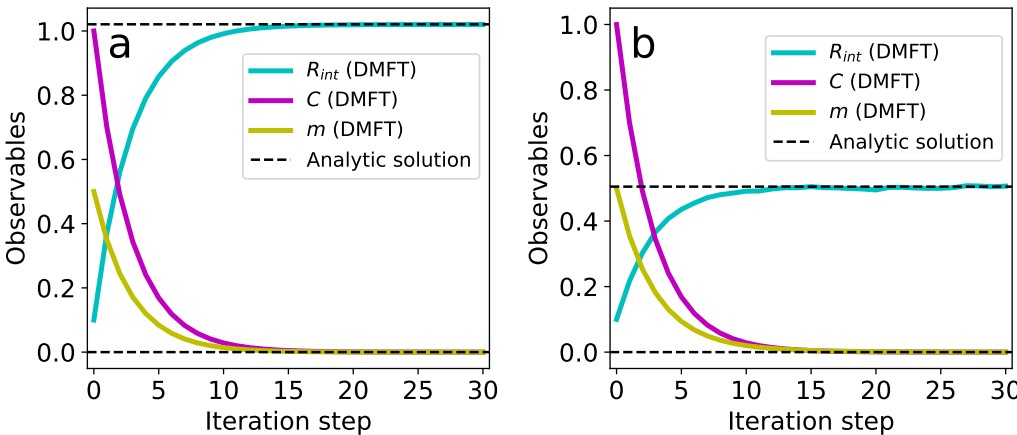

Figure 2: Convergence of observables to analytic fixed point solutions during iteration of the DMFT equation. The parameters set $\{g, \eta, \sigma, \Delta t\}$ is $\{0.2, 0.5, 0.1, 0.1(ms)\}$. (a) Transfer function $\phi = \tanh$. (b) Transfer function $\phi = \text{ReLU}$.

Note that we discard the other root for $R$ because of the divergence in the limit $\eta \to 0$ (keeping $g$ finite). Equation (61) implies that $g^2/(2(1-w)^2) \neq 1$ and $1 - 2g^2\eta > 0$ (see Appendix C).

We remark that we consider here only the trivial fixed point (null activity) and its stability. The analysis of other types of solutions (especially those time-dependent mean-field states) becomes complicated, as shown by random neural networks with i.i.d. couplings in a previous work [12]. In general, we have not a closed-form solution for the integro-differential equations [see Eq. (35) and Eq. (36)]. For the nonequilibrium relaxation of the spherical spin-glass model, an analytic solution can be derived [36].

We compare the fixed point solution obtained directly from Eq. (51) to the analytic fixed point given by Eqs. (60, 56). The fixed point iteration becomes difficult when $\phi = \tanh$ in the current model. This fixed point is a well-known result of $m = C = 0$, which is hard to achieve numerically, because numerical errors always make it impossible to get a perfect zero value. We must set a very small value for the initialization of $C$, or we can just set $C$ to zero and then get $R_{\text{int}}$. In spite of this numerical error, we observe a perfect match (Figure 2) as the iteration step of the DMFT equation [Eq. (51)] increases.

## 4.3 Fluctuation-dissipation theorem

The fluctuation-dissipation theorem (FDT) relates the linear response function to the correlation function in equilibrium, which establishes a model independent relationship connecting the statistics of spontaneous fluctuation to the response to perturbations [30]. A static counterpart is the linear response theory that relates the fluctuation and susceptibility. FDT allows one to predict the mean response to external perturbations without applying any perturbation, and instead, by analyzing the time-dependent correlations. FDT holds particularly in a stochastic system subject to conservative forces and the dynamics bears an equilibrium state [26, 30]. We discuss the relevance of FDT in the context of random neural networks in this subsection.

As we know, dynamical systems tend to be more difficult to study than equilibrium systems, because we have no prior knowledge of the steady state (if any, especially for those nonequilibrium ones) in a general context. In the simplest case, we consider an over-damped Langevin dynamics,

$$\lambda \dot{x}_i(t) = -\frac{\partial \mathcal{H}(\boldsymbol{x})}{\partial x_i(t)} + \eta_i(t), \tag{62}$$

where $\lambda$ is a friction coefficient, $\eta_i(t)$ is a Gaussian white noise, and $\langle \eta_i(t)\eta_j(t') \rangle = 2T\lambda\delta_{ij}\delta(t-t')$. The temperature bridges the relationship between the noise strength and the friction in the equilibrium state. Here, we assume that the dynamics could be interpreted as moving particles in a potential $\mathcal{H}(\boldsymbol{x})$ (or gradient dynamics). This Langevin dynamics, in the long time limit, is able to reach the thermal equilibrium that is captured by the Gibbs-Boltzmann distribution,

$$P(\boldsymbol{x}) \sim \exp\left(-\frac{\mathcal{H}(\boldsymbol{x})}{T}\right), \tag{63}$$

where the Hamiltonian is exactly the potential function that drives the dynamics through Eq. (62). This precise probability measure can be derived from the Fokker-Planck equation by setting the probability current to zero [30]. We have assumed $k_B = 1$ without loss of generality.

Next, we consider a linear and full-symmetric network whose dynamics is governed by

$$\frac{dx_i(t)}{dt} = -x_i(t) + g\sum_{j=1}^{N} J_{ij}x_j(t) + \sigma\xi_i(t), \tag{64}$$

where $J_{ij} = J_{ji}$. By comparing this equation with the Langevin dynamics Eq. (62), we can directly write down the Hamiltonian $\mathcal{H}(\boldsymbol{x}) = -\frac{1}{2}\sum_i x_i^2 - \frac{1}{2}g\sum_{i\neq j} J_{ij}x_i x_j$, and the temperature is determined by $T = \sigma^2/2$. We remark that non-gradient dynamics (e.g., $\eta \neq 1$) may have a non-equilibrium steady state for which FDT breaks. In this simple gradient dynamics, the equilibrium can be reached and the FDT holds as follows,

$$\chi(t,t') = -\frac{1}{T}\partial_t \Delta(t,t')\Theta(t-t'), \tag{65}$$

where the instantaneous response function $\chi(t,t') = \frac{1}{N}\sum_i \frac{\partial \langle x_i(t)\rangle_\xi}{\partial j_i(t)}$ and the time-dependent fluctuation $\Delta(t,t') = \frac{1}{N}\sum_i \langle x_i(t)x_i(t')\rangle_\xi$. These functions also bear the time-translation invariance due to the steady state condition. In addition, we can prove that FDT is valid in the dynamical system of Eq. (64), and we leave the proof to the Appendix D.

An experimentally measurable quantity is the integrated response function calculated by

$$\chi_{\text{int}}(t,t') = \int_{t'}^{t} ds\, \chi(s,t'). \tag{66}$$

Then we rescale the integrated response by the equal time correlation function, and get

$$\hat{\chi}_{\text{int}}(t,t') = \chi_{\text{int}}(t,t')/\Delta(t',t'), \qquad \hat{\Delta}(t,t') = \Delta(t,t')/\Delta(t',t'). \tag{67}$$

Thus, when $t \geq t'$, we have the relation as,

$$\hat{\chi}_{\text{int}}(t,t') = \frac{1}{T}\big(1 - \hat{\Delta}(t,t')\big). \tag{68}$$

Equation (68) establishes an easy way to measure the temperature determined by the slope of the parametric plot $\hat{\chi}_{\text{int}}(t+t_w, t_w)$ versus $\hat{\Delta}(t+t_w, t_w)$ where $t_w$ is a waiting time for reaching the steady state. This temperature is called the effective temperature [32], which may be a constant or changes with the time difference $t$. If the Gibbs-Boltzmann measure exists, the effective temperature coincides with the thermodynamic temperature. However, these two temperatures are not equal in general. Even in aging systems where the the decay of the correlation and response functions depend on the waiting time (how long the system is prepared), the effective temperature may be different for different ranges of the correlation function (displaying multiple relaxation time scales as in mean-field glass models [32]).

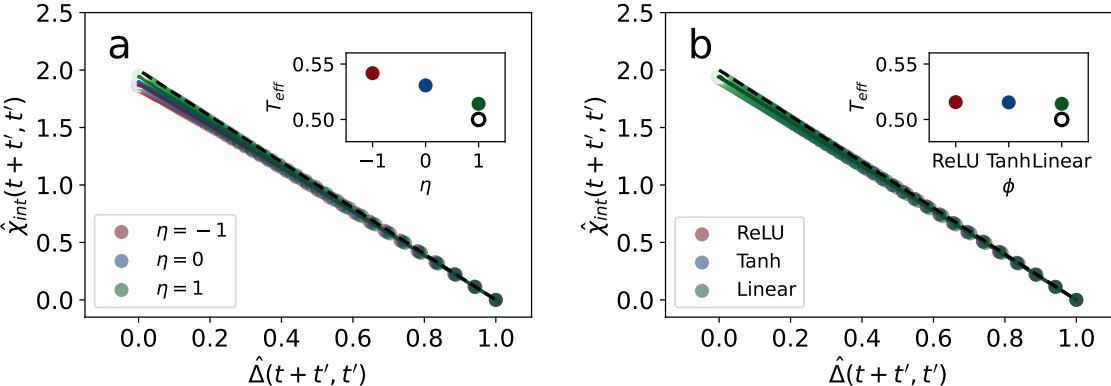

Figure 3: Effective temperature of the system given different asymmetry correlation levels and nonlinearities. The waiting time $t'$ is fixed at 9ms and the color of points becomes lighter as $t$ increases. The dash line indicates FDT for the linear system with symmetric connection $\eta = 1$, whose thermodynamic temperature is $T = \sigma^2/2 = 0.5$ (indicated by a black circle in the inset). (a) Comparison among different asymmetry correlation levels in the linear system. For $\eta = [-1, 0, 1]$, the effective temperatures obtained by a linear fitting are $T_{\text{eff}} = [0.541, 0.531, 0.514]$, respectively (inset). (b) Comparison among different nonlinear functions when $\eta = 1$. For $\phi$ selected to be ReLU, Tanh and linear, the effective temperatures obtained by a linear fitting are $T_{\text{eff}} = [0.516, 0.515, 0.514]$, respectively (inset).

The concept of effective temperature captures the information on the presence of different relevant time scales in the system dynamics [37,38]. The effective temperature may be related to the complexity of an equivalent or approximate potential underlying the dynamics [39], and is measurable from the ratio between spontaneous fluctuation and linear response. For example, aging systems relaxing very slowly could be characterized by an effective thermodynamic behavior where an effective temperature could be computed [32].

A recent work explored the FDT in gradient dynamics of supervised machine learning [17]. Here, we verify the fluctuation-dissipation theorem in the rate model via solving the DMFT equation and provide insights on the long time behavior. We focus on the influence of the nonlinear function and asymmetric connections on the FDT. The results are summarized in Figure (3). For the case where $\eta = 1$ and $\phi$ is linear, the FDT must be valid (see the Appendix D), the slope obtained by the linear fitting indicates a temperature closer to the ground truth compared to other non-gradient dynamics. The slight deviation is caused by the numerical errors of finite discretization time step $\Delta t$ from simulations. Figure 4 shows that the estimated effective temperature for $\eta = 1$ and non-linear systems becomes stable for a large waiting time, but less accurate for asymmetric linear networks due to limited data points for fitting if a large waiting time is considered. Therefore, we set the waiting time to 9ms in our simulations, where the time translation invariance is satisfied (see Figure 5). Interestingly, the nonlinear function and the other asymmetry correlation levels do not yield a strongly-violated FDT, because a linear fitting with a larger effective temperature is observed. This suggests that the out-of-equilibrium steady state with unknown probability measure may be approximated by an equilibrium FDT with a larger effective temperature, which offers an interesting perspective to bridge the non-gradient dynamics (commonly observed in recurrent neural networks) to an effective thermodynamic behavior.

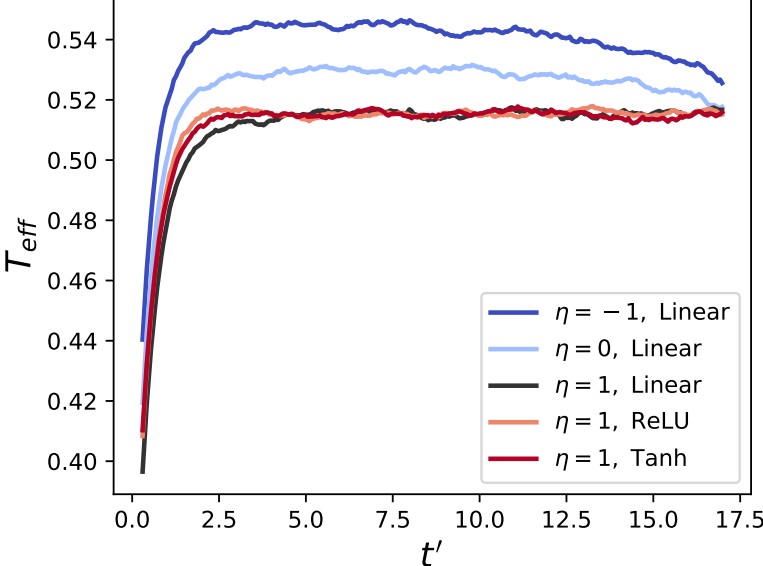

Figure 4: The estimated effective temperature versus the waiting time. Simulation parameters are the same as in Fig. 3.

## 5  Conclusion

In this lecture note, we briefly introduce the path integral framework, from which the dynamical mean-field theory of the stochastic dynamics in high dimensional systems is derived. We also introduce a complementary cavity method to derive the exactly same results. Considering the long time limit of the dynamics, we analyze the fixed point solution of the dynamics by a direct deduction of the DMFT equation [21] and by a static cavity analysis [22]. The FDT is also discussed in the context of random neural networks we consider in this note. Based on these theoretical descriptions, it is interesting to detect the fundamental relationship between spontaneous fluctuations and response functions to external perturbations, especially in the gradient dynamics commonly observed in deep learning [4,17]. The fluctuation dissipation relation is also studied in a recent work for spiking neural networks [40], and the path integral framework can also find applications in revealing inner workings in recurrent network models of memory and decision making [41,42], in low-rank recurrent neural networks [27], and in recurrent neural networks with gating mechanisms [28]. We hope this tutorial will expand the cutting-edge researches of learning in neural networks, inspiring more fundamental physics laws governing high dimensional complex neural dynamics and novel algorithmic designs.

## Acknowledgments

We thank all PMI members for inspiring discussions of the dynamical mean-field theory and path integral formalism.

**Funding information**   This research was supported by the National Natural Science Foundation of China for Grant number 12122515, and Guangdong Provincial Key Laboratory of Magnetoelectric Physics and Devices (No. 2022B1212010008), and Guangdong Basic and Applied Basic Research Foundation (Grant No. 2023B1515040023).

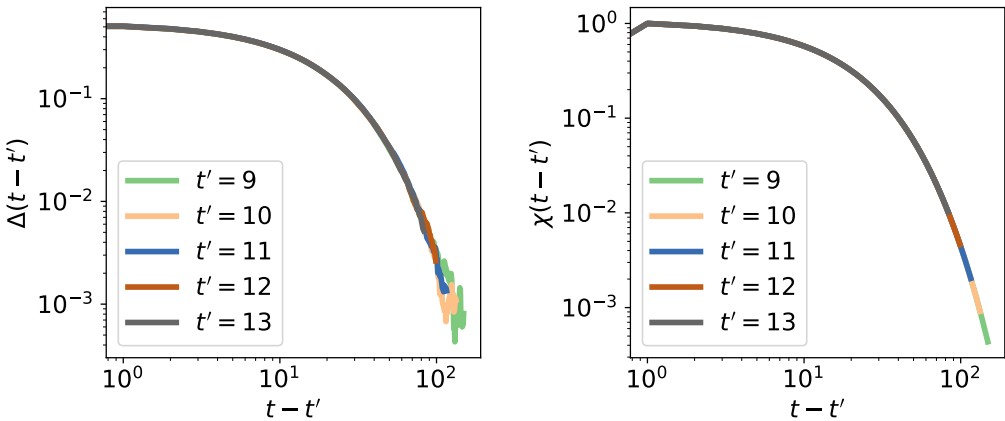

Figure 5: Response and correlation functions depend only on the time difference. Linear dynamics with $\eta = 1$ is considered for an example. Simulation parameters are the same as in Fig. 4.

## A  Novikov's theorem

Novikov's theorem characterizes a useful identity to estimate the dynamic response function. In this section, we give a derivation for both the original rate model Eq.(1) as well as the DMFT equation [Eq. (32)]. We consider a general form of dynamical equations,

$$\dot{x}_i(t) = F_i(\boldsymbol{x}) + c\Xi_i(t) + j_i(t), \qquad i = 1, 2, \ldots, N, \tag{A.1}$$

where $F$ is a general functional of $\boldsymbol{x}$ and $c$ is a constant. Note that, Novikov's theorem is valid only if we consider a Gaussian noise here. Thus, we set a general Gaussian noise with the covariance structure as $\langle \Xi_i(t)\Xi_j(t') \rangle = D_{ij}(t, t')$. For the above dynamics, the response function is defined by

$$R_{ij}(t, t') = \frac{\delta \langle \phi_i(t) \rangle}{\delta j_j(t')}\bigg|_{j=0}. \tag{A.2}$$

The path average $\langle \mathcal{O} \rangle$ can be defined by $\int \int \mathcal{D}\boldsymbol{x} \mathcal{D}\Xi \mathcal{O} P(\boldsymbol{x}|\Xi)P(\Xi)$, where

$$P(\boldsymbol{x}|\Xi) = \prod_{i,t} \delta(\dot{x}_i(t) - F_i(\boldsymbol{x}) - c\Xi_i(t) - j_i(t)),$$

$$P(\Xi) = \frac{1}{Z_\Xi} \exp\left( -\frac{1}{2} \sum_{i,j} \int \int dt \, dt' \, \Xi_i(t) D_{ij}^{-1}(t, t') \Xi_j(t') \right), \tag{A.3}$$

where $Z_\Xi$ is a normalization constant. With this definition, the response function could therefore be calculated as,

$$
\begin{aligned}
R_{ij}(t, t') &= \frac{\delta}{\delta j_j(t')} \int \int \mathcal{D}\boldsymbol{x} \mathcal{D}\Xi P(\Xi) P(\boldsymbol{x}|\Xi) \phi_i(t) \\
&= \int \int \mathcal{D}\boldsymbol{x} \mathcal{D}\Xi \phi_i(t) P(\Xi) \frac{\delta}{c\delta\Xi_j(t')} P(\boldsymbol{x}|\Xi) \\
&= -\int \int \mathcal{D}\boldsymbol{x} \mathcal{D}\Xi \phi_i(t) P(\boldsymbol{x}|\Xi) \frac{\delta}{c\delta\Xi_j(t')} P(\Xi)
\end{aligned}
\tag{A.4}
$$

$$= \frac{1}{c} \int \int \mathcal{D}\boldsymbol{x}\, \mathcal{D}\Xi\, \phi_i(t) P(\boldsymbol{x}|\Xi) \left( \int \sum_k \mathrm{d}s\, D_{jk}^{-1}\left(t',s\right)\Xi_k(s) \right) P(\Xi)$$

$$= \frac{1}{c} \left\langle \phi_i(t) \left( \int \sum_k \mathrm{d}s\, D_{jk}^{-1}\left(t',s\right)\Xi_k(s) \right) \right\rangle,$$

where we have used the property that $P(\boldsymbol{x}|\Xi)$ is symmetric with respect to $\Xi_j(t')$ and $j_j(t')$, and we have applied the integral by parts in the third equality. For the original $N$-neuron model [Eq.(1)], $c = \sigma$, $\Xi_i(t) = \xi_i(t)$ and $D_{ij}(t,t') = \delta_{ij}\delta(t-t')$. We thus have,

$$R_{ij}(t,t') = \frac{1}{\sigma} \left\langle \phi_i(t) \left( \int \sum_k \mathrm{d}s\, \delta_{jk}^{-1} \delta^{-1}(t'-s)\xi_k(s) \right) \right\rangle = \frac{1}{\sigma} \left\langle \phi_i(t)\xi_j(t') \right\rangle, \tag{A.5}$$

where we have used the identity $\int \delta^{-1}(t-s)\delta(s-t')\,\mathrm{d}s = \delta(t-t')$. For the DMFT equation [Eq.(32)] of one-neuron system, $c = 1$, $\Xi(t) = \gamma(t)$ and $D(t,t') = \Gamma(t,t') = g^2 C(t,t') + \sigma^2 \delta(t-t')$. Therefore, the response function becomes,

$$R(t,t') = \left\langle \phi(t) \int \mathrm{d}s\, \Gamma^{-1}(t',s)\gamma(s) \right\rangle. \tag{A.6}$$

## B  Static cavity method for recurrent dynamics

In this section, we introduce the static cavity method [22] to derive the self-consistent equations for the fixed point of dynamics. We consider the rate model with the ReLU transfer function $\phi(x) = x\Theta(x)$. The procedure starts from the fixed-point condition of the noise-free rate model [Eq. (1)], which is

$$x_i = g \sum_{j=1}^{N} J_{ij} v_j + j_i, \tag{B.1}$$

where we set $v_j = \phi(x_j)$ for simplicity. Note that asymmetric connections incorporate correlation between $J_{ij}$ and $v_j$, and thereby the central limit theorem in the summation does not apply. To overcome this barrier, we can remove the contribution of $J_{ji}$ and consider this deletion as a small perturbation. Therefore, the sum in Eq. (B.1) can be separated into two parts as,

$$x_i = g \sum_{j=1}^{N} J_{ij} v_{j \to i} + g \sum_{j=1}^{N} J_{ij} \delta v_j + j_i, \tag{B.2}$$

where $v_{j \to i}$ denotes the firing rate of neuron $j$ in the absence of $J_{ji}$, and $\delta v_j$ is the perturbation caused by the presence of $J_{ji}$. According to the central limit theorem, the first term on the right hand side can now be treated as a Gaussian field, which we denote as $\tilde{\gamma}_i$ and compute the variance,

$$\langle (\tilde{\gamma}_i)^2 \rangle = g^2 \tilde{C}, \tag{B.3}$$

where $\tilde{\phantom{x}}$ indicates the cavity quantity. Note that, $\tilde{C} = \langle v_{j \to i}^2 \rangle$ is the self-consistent cavity variance function. As for the second term, we compute the perturbation by a linear response approximation,

$$\delta v_j = \sum_{k=1}^{N} R_{jk} \eta_k, \tag{B.4}$$

where $R_{jk} = \frac{\delta v_j}{\delta j_k}$ is the linear response function. The small perturbation $\eta_k$ here is actually the contribution from neuron $i$ to neuron $k$ through $J_{ki}$, which is exactly $J_{ki} v_i$. Therefore, the

second term in the r.h.s. of Eq. (B.2) becomes,

$$g \sum_{j=1}^{N} J_{ij} \delta v_j = g \sum_{j=1}^{N} \sum_{k=1}^{N} J_{ij} R_{jk} J_{ki} v_i$$
$$\approx g \eta^2 R v_i, \tag{B.5}$$

where $R = \frac{1}{N} \sum_j R_{jj}$ . Note that, the approximation in the last equality is exactly the same as what we did in the dynamical cavity approach [see Eqs.(42,43)].

Finally, we recast the fixed point equation ($j_i = 0$) as

$$x_i = \tilde{\gamma}_i + w v_i, \tag{B.6}$$

where $w = g \eta^2 R$ and $\langle \tilde{\gamma}_i^2 \rangle = g^2 \tilde{C}$. The above equation can be transformed to $v_i = \phi(\tilde{\gamma}_i + w v_i)$ and then written as a function of $\tilde{\gamma}_i$ as,

$$v_i = \frac{\tilde{\gamma}_i \Theta(\tilde{\gamma}_i)}{1 - w} \equiv \psi(\tilde{\gamma}_i). \tag{B.7}$$

We then compute the cavity variance function and the linear response function as follows,

$$\tilde{C} = \langle v_i^2 \rangle = \int \left( \frac{\tilde{\gamma}_i \Theta(\tilde{\gamma}_i)}{1 - w} \right)^2 p(\tilde{\gamma}_i) d\tilde{\gamma}_i = \frac{g^2 \tilde{C}}{2(1-w)^2},$$
$$R = \left\langle \frac{\delta v_i}{\delta \tilde{\gamma}_i} \right\rangle = \int \frac{\Theta(\tilde{\gamma}_i)}{1 - w} p(\tilde{\gamma}_i) d\tilde{\gamma}_i = \frac{1}{2(1-w)}. \tag{B.8}$$

We remark that $\tilde{C}$, under the limit of $N \to \infty$, can be replaced by the full variance function $\tilde{C} = \langle v_i^2 \rangle$. The above results are now exactly the same with Eq. (60).

## C  Stability analysis for the ReLU transfer function

The stability of the trivial fixed point can be linked to the connectivity spectrum. To be more precise, we take the rate model whose transfer function is $\phi = \tanh$ as an example. We can first linearize the neural dynamics Eq. (1) (in the absence of white noise) around the fixed point ($\boldsymbol{x}^* = 0$),

$$\dot{\Delta x_i}(t) = \sum_j D_{ij} \Delta x_j(t) \quad \rightarrow \quad \Delta \boldsymbol{x}(t) = \exp(Dt) \Delta \boldsymbol{x}(0), \tag{C.1}$$

where

$$D_{ij} = -\delta_{ij} + g J_{ij} \phi'(x_j^*) = -\delta_{ij} + g J_{ij}, \tag{C.2}$$

is the local Jacobian matrix at the fixed point. The eigenvalues of this Jacobian matrix determine the stability of the local dynamics. In particular, if the eigenvalues with the largest real part cross zero along the real axis, the dynamics becomes chaotic in our current model. In general, the instability is not a sufficient but necessary condition for the transition to chaos [7]. Moreover, the spectrum of $J_{ij}$ is actually a well-known elliptic law [43],

$$\rho(\lambda) = \begin{cases} \frac{1}{\pi(1-\eta^2)}, & \left(\frac{x}{1+\eta}\right)^2 + \left(\frac{y}{1-\eta}\right)^2 < 1, \\ 0, & \text{otherwise}, \end{cases} \tag{C.3}$$

where $\lambda$ is the eigenvalue of complex value, while $x$ and $y$ are the coordinates on the real- and imaginary-axis. The special case of $\eta = 0$ gives the circular law in random matrix theory.

Thus, the eigenvalue with the largest real part of $D_{ij}$ is $g(1+\eta)-1$. Consequently, the stability condition is given by $g(1+\eta) < 1$.

However, the Jacobian is ill-defined when $\phi = \text{ReLU}$. Instead, we rely on the static cavity method [22]. For the sake of clarity, we follow the same notations introduced in Appendix *B*. Our starting point is the relation of $v_j = \psi(\tilde{\gamma}_j)$, which states that the firing-rate of neuron $j$ could be seen as a function of its cavity input. From the cavity idea, the presence of neuron $i$ contributes to a perturbation $\delta v_j$, i.e.,

$$\delta v_j = \psi'(\tilde{\gamma}_j)\left[ g\sum_{k\neq i} J_{jk}\delta v_k + gJ_{ji}v_i \right], \tag{C.4}$$

where a linear expansion at $\tilde{\gamma}_j$ is used when the effect of the cavity operation is small. Note that the term in the bracket denotes the deviation of the cavity input to neuron $j$ between with and without the neuron $i$. It is reasonable that if the fixed point is stable, the variance of $\delta v_j$ must be finite and positive [22]. Taking the average over the network statistics, we get $\langle \delta v_j \rangle = 0$. To compute the variance, we square both sides of Eq. (C.4) and take the disorder average, which results in,

$$\langle (\delta v)^2 \rangle = g^2 \chi_\phi \langle (\delta v)^2 \rangle + \frac{g^2}{N}\chi_\phi v_i^2, \tag{C.5}$$

where $\chi_\phi = \left\langle (\psi'(\tilde{\gamma}_j))^2 \right\rangle$. This equation leads to,

$$N\langle (\delta v)^2 \rangle = \frac{g^2\chi_\phi}{1 - g^2\chi_\phi}v_i^2. \tag{C.6}$$

To ensure $\langle (\delta v)^2 \rangle$ physical, the condition $1 - g^2\chi_\phi > 0$ must be satisfied. To proceed, we first compute $\chi_\phi$,

$$\chi_\phi = \left\langle (\psi'(\tilde{\gamma}_j))^2 \right\rangle = \int \left( \frac{\Theta(\tilde{\gamma}_i)}{1-w} \right)^2 p(\tilde{\gamma}_i)d\tilde{\gamma}_i = \frac{1}{2(1-w)^2}. \tag{C.7}$$

Note that $w = g^2\eta R$ and $R = \frac{1}{2(1-w)}$, which implies that

$$g^2 R\eta = \frac{1-\Delta}{2}, \tag{C.8}$$

where $\Delta = \sqrt{1 - 2g^2\eta}$. The stability thus requires that $\frac{g^2}{2(1-w)^2} < 1$, which finally leads to the condition by using Eq. (C.8),

$$g(1+\eta) < \sqrt{2}. \tag{C.9}$$

The stability condition further implies that $1 - 2g^2\eta > 1 - 2g^2(\sqrt{2}/g - 1) \geq 0$.

# D   Derivation of fluctuation-dissipation theorem in equilibrium

In this section, we give a proof of fluctuation-dissipation theorem for the model [Eq. (1)] with linear transfer function and fully-symmetric connections. We begin the derivation by rewriting the model in the vector form,

$$\dot{x}(t) = -x(t) + Jx(t) + \sigma\xi(t) + j(t), \tag{D.1}$$

where $\boldsymbol{J} = \boldsymbol{J}^T$ and $\langle \xi(t)^T \xi(t') \rangle = \mathbb{1}_{N \times N} \delta(t - t')$ by construction. The solution of this linear dynamics can be given by,

$$\boldsymbol{x}(t) = \int_0^t dt' e^{(-1+\boldsymbol{J})(t-t')} \left[ \sigma \xi(t') + \boldsymbol{j}(t') \right]. \tag{D.2}$$

From this solution, we can compute the response function [44],

$$\boldsymbol{\chi}(t, t') = \left. \frac{\partial \langle \boldsymbol{x}(t) \rangle}{\partial \boldsymbol{j}(t')} \right|_{\boldsymbol{j}=0} = \Theta(t - t') e^{(-1+\boldsymbol{J})(t-t')}, \tag{D.3}$$

as well as the correlation function,

$$\begin{aligned}
\boldsymbol{\Delta}(t, t') &= \left\langle \boldsymbol{x}(t) \boldsymbol{x}^{\mathrm{T}}(t') \right\rangle \\
&= \sigma^2 \int_0^t \int_0^{t'} dt'' dt''' e^{(-1+\boldsymbol{J})(t-t'')} \left\langle \xi(t'') \xi^{\mathrm{T}}(t''') \right\rangle e^{(-1+\boldsymbol{J}^{\mathrm{T}})(t'-t''')} \\
&= \sigma^2 \int_0^{\min(t,t')} dt'' e^{(-1+\boldsymbol{J})(t-t'')} e^{(-1+\boldsymbol{J}^{\mathrm{T}})(t'-t'')} \\
&= \sigma^2 \int_0^{\min(t,t')} dt'' e^{(-1+\boldsymbol{J})(t+t'-2t'')},
\end{aligned} \tag{D.4}$$

where the bold functions $\boldsymbol{\chi}(t, t')$ and $\boldsymbol{C}(t, t')$ refers to $N \times N$ matrices. Next, we calculate the mean auto-correlation function [44],

$$\begin{aligned}
\Delta(t, t') &\equiv \frac{1}{N} \operatorname{Tr} \boldsymbol{\Delta}(t, t') \\
&= \sigma^2 \int_{-2}^2 \frac{dk}{2\pi} \sqrt{4 - k^2} \int_0^{\min(t,t')} dt'' e^{(-1+k)(t+t'-2t'')} \\
&= \sigma^2 \int_0^{\min(t,t')} dt'' \frac{I_1(2(t + t' - 2t''))}{t + t' - 2t''} e^{-(t+t'-2t'')} \\
&= \sigma^2 \int_{|t-t'|}^{t+t'} dw \frac{I_1(2w)}{2w} e^{-w},
\end{aligned} \tag{D.5}$$

where the trace used in the second line can be seen as an integral over the eigenvalues. Wigner semi-circle law is used here for the eigenvalue spectrum of fully-symmetric matrix $\boldsymbol{J}$ [29]. In the second line, a substitution $k = 2\cos\theta$ is used and the modified Bessel function of the first kind is introduced, i.e., $\frac{I_1(\tau)}{\tau} = \frac{1}{\pi} \int_0^\pi d\theta (\sin\theta)^2 e^{\tau \cos\theta}$. The final step also involves a substitution of $w = t + t' - 2t''$. In the long time limit, the upper limit of integral tends to $\infty$ and the mean auto-correlation function becomes time translation invariant.

Follow the same procedure, we can compute the mean response function,

$$\chi(t, t') \equiv \frac{1}{N} \operatorname{Tr} \boldsymbol{\chi}(t, t') = \Theta(t - t') \frac{I_1\left(2\left(t - t'\right)\right)}{t - t'} e^{-(t-t')}. \tag{D.6}$$

Finally, it would be easy to verify the following FDT using Eq. (D.4) and Eq. (D.5),

$$\chi(t, t') = -\frac{1}{T} \partial_t \Delta(t, t') \Theta(t - t'), \tag{D.7}$$

where $T = \sigma^2/2$ represents the thermodynamic temperature. Performing the Fourier transform, we can also recast the FDT in the frequency domain $\Delta(\omega) = \frac{2T}{\omega} \operatorname{Im} \chi(\omega)$.

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
