# Peer review of "Introduction to dynamical mean-field theory of randomly connected neural networks with bidirectionally correlated couplings"

_SciPost Physics Lecture Notes, doi:SciPost Phys. Lect. Notes 79 (2024)_

## Round 3 · Referee Report · Anonymous (Referee 2) · 2024-1-20

Report

My main concerns have been addressed by the authors response.

---

## Round 3 · Author Response

We first thank the referees for their efforts to evaluate our manuscript. The comments are valuable to enhance the quality of the manuscript. We now provide point-by-point responses to all comments, and a list of major changes is finally appended. Responses to the first referee's comments: (1) The topics reviewed by the Authors have already been largely covered in previous literature, notably in references [11,21,22,23,24]. I find that the manuscript fails to contextualize appropriately its specific contribution and goals with respect to previous contributions.

ANS: We apology for the unclear motivation. In the revised version, we clearly comment on the motivation and relationship with previous works (see the added last paragraph in the introduction, page 2-3, and the added second paragraph in page 15). We hope the motivation is now clear.

(2) The paper contains significant flaws in the presentation. My major concerns are: (i) Previous literature is cited inappropriately and sometimes overlooked, both regarding original work and review papers. (ii) Some technical derivations and comment of the results lack clarity.

ANS: Thank you for providing us your criticism, which are addressed in our new version of the manuscript. Please see the following point-to-point reply.

(3) The Authors should deemphasize the generality of their derivations. "Generic random neural network" (Sec. 2) is too broad with respect to the actual model studied here. E.g., interactions J_ij (i<j) are iid, while a more generic covariance structure could be considered, as well as a more complicated noise structure or interactions (see [27] and missing references [d,e]).

ANS: Thanks for the comment. We remove the word “generic”, and instead, we write more specifically with networks with correlated bidirectional synapses, and further comment on other types of synaptic structures below Eq. (2).

(4) The introduction fails to correctly refer to previous review literature: [21] and [23] are only introduced as references in the technical section, while these works already cover most of the topics of the manuscript and therefore should be discussed in the introduction.

ANS: Thank you very much for raising this missing point. We address this point using a separate paragraph in the introduction (see the last paragraph).

(5) Missing reference [a] (see Chapt. VI) should be cited when introducing the dynamical cavity method . Similarly, I think that [a] is a more appropriate citation for "the replica analysis in equilibrium spin glass theory" (page 8, below Eq. 25) .

ANS: Thank you very much for raising this issue. We have now cited this classic book in both places.

(6) The Authors should clarify a crucial difference between their derivation and other learning theory references such as [14,27], i.e., in the latter the interaction weights are learned.

ANS: Thank you very much for raising this point. We clarify this point as “In particular, a dynamical partition function was derived for dynamical weight variables in supervised learning, from which the correlation and response functions are computed [17], while a recent work derived dynamical field theory for infinitely wide neural networks trained with gradient flow, at an expensive computational cost [15]. We will next provide a detailed explanation of this field-theoretical formalism, yet in random neural networks with predesigned synaptic structures.” in the second paragraph of the introduction.

(7) I find the discussion confusing and sloppy regarding: the difference between relaxation to equilibrium and non-equilibrium steady state; the closure of the dynamical equations on fixed point solutions and when this connection is missing or only partially understood (e.g., glassy systems), the physical meaning of (effective) FDT. In particular, the following sentences are not precise: Page 8: "This relation bears the similarity with the linear response relation in equilibrium statistical physics" Page 9: "Note that the two point correlation could relax to the EA order parameter" - Missing reference: [b] for the numerical integration of the DMFT self-consistent process.

ANS: Thank you very much for raising these confusing points. In page 9, we remove the sentence. In page 12, we rewrite the sentence as “Note that in mean-field spin glass models [32], the long time-difference limit of the two-point correlation corresponds to the Edwards-Anderson order parameter”. We also cite the missing reference [33].

We distinguish the equilibrium from the non-equilibrium steady state in the revised version, see section 4.3, e.g., if equilibrium can be achieved, “steady” is replaced by “equilibrium”.

On the fixed point analysis, we add a separate paragraph in page 15, “We remark that we consider here only the trivial fixed point (null activity) and its stability. The analysis of other types of solutions (especially those time-dependent mean-field states becomes complicated, as shown by random neural networks with i.i.d. couplings in a previous work [12]. In general, we have not a closed-form solution for the integro-differential equations [see Eq. (35) and Eq. (36)]. For the nonequilibrium relaxation of the spherical spin-glass model, an analytic solution can be derived [36].”

On the physical meaning of effective FDT, we also add a separate paragraph in page 17, as “The concept of effective temperature captures the information on the presence of different relevant time scales in the system dynamics [37,38]. The effective temperature may be related to the complexity of an equivalent or approximate potential underlying the dynamics [39], and is measurable from the ratio between spontaneous fluctuation and linear response. For example, aging systems relaxing very slowly could be characterized by an effective thermodynamic behavior where an effective temperature could be computed [32].”

(8) The gap between the measured temperature and the noise correlation when \eta=1 suggests that there is a problem in the extrapolation of Te. These two quantities should match. Is this value of Te stable when increasing t' in Fig.3 ? The authors should clarify this extrapolation procedure: they say it is due to "numerical errors by simulations", but aren't these numbers extrapolated from DMFT? I don't understand the need to specify that "time is measured in units of milliseconds" in this case. Missing reference: [c] where a very similar analysis of effective temperature was carried out for the stochastic gradient descent algorithm.

ANS: Thank you very much for raising this point. We interpret the numerical errors as the finite time step for discretization, since what we solve is the integro-differential equations. Decreasing the discretization time step (0.06 compared to previous 0.1), the new simulation results show that the gap between the DMFT and ground truth temperatures is minimized.

Figure 4 shows that the estimated effective temperature for η = 1 and non-linear systems becomes stable for a large waiting time, but less accurate for asymmetric linear networks due to limited data points for fitting if a large waiting time is considered. Therefore, we set the waiting time to 9ms in our simulations, where the time translation invariance is satisfied (see Figure 5) .

Choosing the unite of milliseconds is common in simulating neural dynamics in brain circuits (e.g., time scales for rate dynamics can be considered, see also the book Ref. 2). The other advantage is that it would be helpful to check time translation invariance. We add the above comments in our main text (see the last paragraph of sec 4.3).

We cite this reference as “A recent work explored the FDT in gradient dynamics of supervised machine learning [17].” in the last paragraph of sec 4.3.

(9) The Authors choose to adopt Ito's convention, which leads to convenient simplifications in the computation. This choice should be remarked in a clearer way.

ANS: Thank you very much for raising this point. We clarify this point as “There are two types of discretization schemes (Ito versus Stratonovich convention). The Ito one is simpler because the equal time response function vanishes. Note that for different times, the two discretization schemes are equivalent [25]. ” below Eq. (5).

(10) There is a formatting problem on Page 8.

ANS: We are not sure which formatting problem, but we may solve this in the revised version.

Overall, we are grateful to you for your useful comments that would definitely improve the quality of our manuscript. In addition, all missing references are now cited.

Responses to the second referee's comments: (1) The presentation of the DMFT method in simple models of recurrent neural networks is very pedagogical and sufficiently concise to condense the main points in a few pages.

ANS: We appreciate your highly positive evaluation of the quality of this tutorial. We have further improved the quality following your and the other referee’s advices.

(2) I believe that the bibliography could be improved and the introduction could be expanded.

ANS: The introduction is greatly expanded in the revised manuscript, and 12 additional references are added (see the list of major changes).

(3) In equation[6] the noise term is evaluated at time t. However, since the authors are using the Ito convention, shouldn't it be evaluated at time t-1 under the convention that this noise is independent of x[t-1].

ANS: Thank you very much for this correction. We have corrected this typo [see Eq.(6) and Eq. (7)].

(4) In sec. 3.2 one could also make reference to chapter 6 of the book by Mézard, Parisi and Virasoro (spin glass theory and beyond), where the very same line of reasoning on a very similar model was presented.

ANS: We have put this citation in the revised version.

(5) Maybe one can also comment that a damping term is sometimes useful to help the convergence of the kernels in the numerical algorithm to solve the DMFT equations (see section 4.1). One can also add that running several stochastic trajectories is an operation that is easily done in parallel.

ANS: Thank you very much for this nice comment. We have added your insights below the numerical scheme, as “We remark that a damping term would be helpful to speed up the convergence and running in parallel several stochastic trajectories is also a useful strategy.”

(6) It may be useful to discuss how the response function is computed in the numerical simulations used in Fig.1 to compare with the numerical solution of the DMFT equations.

ANS: Thank you very much for this suggestion. We add one sentence, “We used Novikov’s theorem to compute the response function for this example (see Eq. (73) in Appendix A).” in the first paragraph of page 13.

Overall, we are grateful to the referee for providing us many useful suggestions that finally improve the quality of the paper.

---

## Round 3 · List of Changes

List of major changes: 1. The title is changed by replacing “generic” with a more specific term, and we also correspondingly rephrase this word in the main text. 2. In the second paragraph of the introduction, we highlight the difference of our focus from those recent works applying dynamical mean-field theory to learning. 3. We cite the original book “spin glass theory and beyond” in the third paragraph when introducing the dynamic cavity method, and in Sec 3.2 as well. 4. In the last paragraph (newly added) of the introduction, we remark the motivation of this short tutorial, and differences from previous reviews. One additional review paper (Ref 24) is cited. 5. We add a comment below Eq. (2), “Note that other types of network structures can also be similarly treated with the following MSRDJ formalism, e.g., the low-rank random recurrent networks [27], and the random recurrent networks with gating mechanisms [28].” 6. The last term in Eq. (3) is redundant and thus deleted. 7. We add notes about Ito and Stratonovich convention below Eq. (5). 8. We delete the sentence “this relation bears the similarity with ….statistical physics” below Eq. (29). 9. Below Eq. (32), we add a comment about the local chaos hypothesis studied in the paper by B. Cessac (1995). 10. We rewrite the EA order parameter sentence in the first paragraph of Sec 4.1, and cite the paper by M Opper et.al., in 1992. We also add a simulation remark below the numerical scheme (Sec 4.1), suggested by the second referee. 11. We add one sentence, “We used Novikov’s theorem to compute the response function for this example (see Eq. (73) in Appendix A).” in the first paragraph of page 13. 12. Just before the last paragraph of Sec 4.2 (page 15), we add a separate paragraph to comment on the fixed-point analysis. 13. We cite the work by P Urbani et.al, in the last paragraph of Sec 4.3. 14. Figure 3 is replaced by a new one since new simulation parameters are used, and now the estimated effective temperature gets closer to the true one. Figure 4 and Figure 5 are also added and discussed in the last paragraph of Sec 4.3. 15. We replace “steady” by “equilibrium” in the first paragraph of Sec 4.3 where necessary. 16. We add further comments on the effective temperature in a separate paragraph just before the last paragraph of sec 4.3 (page 17). 17. References 3, 18, 20, 24, 27-28, 31, 33, 36-39 are added.

---

## Editorial Decision

published